# Additive Modulated Perovskite Microstructures for High Performance Photodetectors

**DOI:** 10.3390/mi11121090

**Published:** 2020-12-10

**Authors:** Zhan Gao, Yifan Zheng, Guancheng Huang, Genjie Yang, Xinge Yu, Junsheng Yu

**Affiliations:** 1State Key Laboratory of Electronic Thin Films and Integrated Devices, School of Optoelectronic Science and Engineering, University of Electronic Science and Technology of China (UESTC), Chengdu 610054, China; zhangao2-c@my.cityu.edu.hk (Z.G.); yifanzheng_uestc@163.com (Y.Z.); dkdkdkdz@163.com (G.H.); genjieyang@std.uestc.edu.cn (G.Y.); 2Department of Biomedical Engineering, City University of Hong Kong, Hong Kong 999077, China

**Keywords:** perovskite film, MAPbI_3_ microstructure, additive modulation, photodetectors, high detectivity

## Abstract

Organic-inorganic hybrid perovskites have been widely used as light sensitive components for high-efficient photodetectors due to their superior optoelectronic properties. However, the unwanted crystallographic defects of perovskites typically result in high dark current, and thus limit the performance of the device. Herein, we introduce a simple route of microstructures control in MAPbI_3_ perovskites that associates with introducing an additive of 3,3,4,4-benzophenonetetracarboxylic dianhydridean (BPTCD) for crystallization adjustment of the perovskite film. The BPTCD additive can facilitate the formation of high-quality perovskite film with a compact and nearly pinhole-free morphology. Through characterizing the molecular interactions, it was found that the carbonyl groups in BPTCD is the key reason that promoted the nucleation and crystallization of MAPbI_3_. As a result, we obtained high-efficient and stable perovskite photodetectors with low dark current of 9.98 × 10^−8^ A at −0.5 V, an on/off ratio value of 10^3^, and a high detectivity exceeding 10^12^ Jones over the visible region.

## 1. Introduction

Recently, organic-inorganic hybrid perovskites have attracted enormous interest due to their superior optoelectronic properties, including high charge carrier mobility, long carrier diffusion length, and a broad range of bandgap [1,2,3,4,5,6,7,8,9,10]. These remarkable optoelectronic properties have provoked numerous researches on perovskite electronics, such as solar cells [1,2,3,4], light-emitting diodes [5,6,7], lasers [8,9,10] and photodetectors [11,12,13,14]. Among these devices, perovskite photodetectors (PePDs) can make instantaneously transition between optical and electrical signals, which broaden a route for many important applications including environmental monitoring, optical communication, remote control, and biological sensing [11,12]. In 2014, Hu et al. reported a wide optical range PePD for the first time, whose photo-responsivity was 3.49 A W^−1^, 0.0367 A W^−1^ at 365 nm and 780 nm with a voltage bias of 3 V [11]. Later on, Yang et al. demonstrated a solution-processed PePD based on MAPbI_3-x_Cl_x_ film, with a high detectivity (D*) value of 8 × 10^13^ Jones at −100 mV [12]. Since then, various strategies, including tuning film quality of perovskites [14,15,16,17,18,19,20,21], device structure optimization [22,23,24,25,26,27], and introducing novel materials [28,29,30,31] have been adopted to enhance the device performance.

The performance of PePDs is strongly related to the optoelectronic properties of the perovskite film, which mainly associates with its intrinsic natures such as crystallographic defects, surface morphology, and crystalline size [32,33]. Many routes have been explored to improve the quality of perovskite film. Surface modification of perovskite films is one of the effective methods to fabricate high-qualiy perovskite film [4,13,34,35]. Zheng et al. applied poly(perfluorobutenylvinylether) to modifiy the perovskite surface, obtaining a efficient and stable perovskite solar cell [4]. Ambient Air Plasma was also studied to modify the surface of perovskite film [34]. Zhao et al. used an anti-solvent additive to improve the quality of the perovskite films [13]. The resulted perovskite films exhibit good crystallinity and uniformity, and the correponding PePDs revealed a high D* of 10^12^ and 10^11^ Jones in the Visible and near-infrared regions, respectively. Another successful example is introducing vapor-assisted process for the purpose of perovskite film modification, and the D* of the resulting PePDs could achieve 3 × 10^12^ Jones at −0.1 V [15]. Furthermore, the crystallization of the perovskite films can also be optimized by novel film deposition method, such as nano-imprint lithography [16,17]. Nanoscale-patterned and vertically grown halide perovskites realized by these imprinting methods improved the photo-response and performance of the PePDs significantly. Although the pattern-assisted processes can improve the quality of perovskite film obviously, the requirement of special pattern molds increases the complexity of fabrication processes. Hence, developing a concise and low-cost method for high-quality perovskite can contribute fudamental understanding of high performance PePDs.

In this work, we present a simple approach to tune the MAPbI_3_ microstructures and improve the film quality, by introducing a crystallization controlling additive of 3,3,4,4-benzophenonetetracarboxylic dianhydridean (BPTCD). The BPTCD additive can facile the heterogeneous nucleation of the MAPbI_3_ through the interaction between carbonyl (C=O) and PbI_2_, which lowering the energy barrier of nucleation and thus significantly suppressed the dark current (*I*_d_) in the PePDs. Based on the additive controlled high-quality perovskite film, we realized a high performance PePDs with a low *I*_d_ of 9.98 × 10^−8^ A at −0.5 V, a high on/off ratio of 10^3^, and a D* value 10^12^ Jones over visible region.

## 2. Materials and Methods

The perovskite precursor solution was prepared by mixing 744 mg MAI with 254.3 mg PbI_2_ (Polymer Light Technology Corp., Xi’an, China) in 1 mL DMF (Sigma-Aldrich, St. Louis, MO, USA). BPTCD was purchased from Sigma-Aldrich and used as received. Adding BPTCD into the perovskite precursor formed solution with concentrations ranging from 1 to 4 wt.%. The above solution was stirred at 60 °C overnight to ensure the adequate dissolution. The device fabrication began on an ITO coated glass substrate where thin film ITO with a sheet resistance of 15 Ω/sq acted as anodes. The substrates were cleaned in an ultrasonic bath with detergent water, acetone, deionized water, and isopropyl alcohol successively. Then the ITO glasses were nitrogen blew dry and then treated with oxygen plasma under a pressure of 25 Pa for 5 min to modify the surface energy of the ITO. PEDOT:PSS was spin-coated at 5000 rpm for 60 s and immediately annealed at 145 °C for 15 min. Then, substrates were transferred into a N_2_ filled glove box. The perovskite solution was spin-coated onto the PEDOT:PSS film (2500 rpm for 40 s). 300 μL chloro-benzene (CB) was dropped onto the samples 7 s after the start of spin-coating. Samples were then annealed at 105 °C for 15 min. Afterwards, the dichlorobenzene solvent of PC_61_BM was spun at 3000 rpm for 40 s on the perovskite film and annealed at 100 °C for 20 min. Finally, the devices were completed by consecutive vacuum deposition of an Ag cathode (100 nm) under 10^−5^ mbar. The overlap between ITO and Ag electrodes was 0.02 cm^2^, which is the active absorption area of the devices. For the reference n-type (electron-only) devices, the perovskite film was sandwiched in the middle of the PC_61_BM films. PC_61_BM with a concentration of 20 mg/mL was spun-coated on ITO at a rate of 2000 rpm for 60 s inside the glove box. Finally, Ag (100 nm) is thermally deposited under high-vacuum condition.For the Fourier Transform Infrared (FTIR) measurements, all samples were spin-coated on the ITO coated glass substrates at a rate of 1000 rpm for 60 s. The concentration of the precursor solution was 50 mg/mL for the pure BPTCD sample. The precursor solution for the BPTCD:PbI_2_ sample was prepared by dissolving 25 mg BPTCD and 40 mg PbI_2_ in 1 mL DMF.

The current density-voltage-luminance (J-V-L) characteristics were tested in dark and under a white light (100 mW/cm^2^) with a Keithley 4200 source. Quantum efficiency test system (Zolix SC 100, Beijing, China) was used to obtain the EQE spectra then detectivity was calculated based on the obtained EQE results. The absorption spectra were acquired on a Horiba 320 detector. The surface morphology and the cross-section view of the perovskite film were characterized by scanning electron microcopy (SEM, FEI Inspect F50, FEI Company, Eindhoven, The Netherlands). Surface morphologies of active layers were characterized by atomic force microscope (AFM, AFM 5500, Agilent, Tapping Mode, Chengdu, China). The crystalline structures were characterized by X-ray diffraction (XRD, D2 PHASER, Karlsruhe, Germany). The Thermo Scientific Escalab 250Xi with an ultraviolet photoelectron spectroscopy (UPS) system was used to measure the energy level of the perovskite layers. Fourier-transform infrared (FTIR) measurement was conducted with a FTIR spectrometer (Thermo Cientific, Nicole−10, Waltham, MA, USA). The impedance spectra were measured by Agilent precision impedance analyzer 4294A. All the measurements were performed in air at room temperature without encapsulation.

## 3. Results

Figure 1a shows the explored view of the schematic diagram for the PePDs in this work, with the multiple layers structure of indium tin oxide (ITO)/poly(3,4-ethylenedioxythiophene):polystyrene sulfonate (PEDOT:PSS) (40 nm)/MAPbI_3_:BPTCD (~400 nm)/PC_61_BM (70 nm)/Ag (~100 nm). To investigate the photo responsivity of the perovskite film, we firstly measured the UV-Visible absorption spectra of the perovskite films with and without BTPCD. As shown in Figure 1b, the absorption spectra of these two films are almost overlap, indicating that BPTCD additive doesn’t change the visible light absorption properties of the perovskite film. Microstructures of the MAPbI_3_ were then characterized by XRD measurement and summarized in Figure 1c. The XRD patterns show three main diffraction peaks at 14.3°, 28.6° and 32.2° for all perovskite films, which represent the (110), (220) and (310) planes, respectively. The other weak diffraction peaks appear at 19.9°, 23.3°, 24.5°, 40.5°, 43.1° and 50.2° represent crystal indices (112), (211), (202), (224), (314) and (404) of the perovskites. All the diffraction peaks are in good agreement with literature reports [12,13,36]. Such results indicate that the orthorhombic crystal structure exists in both pure perovskite and the BPTCD added perovskite films [37]. However, the intense diffraction peaks of the BPTCD added perovskite film demonstrate greater crystallinity. We note here that the enhancements of crystallinity are favorable to reduce the traps, suppress the recombination of charge carries and consequently lower the dark current.

Additionally, the pure perovskite film exhibits a weak diffraction peak at 12.5°, which corresponds to the (001) plane of unconverted PbI_2_. The existence of PbI_2_ is typically due to the inefficient reaction between the PbI_2_ and MAI precursor or decomposition of MAPbI_3_ induced by ambient exposure [37,38,39]. Owing to the passivation function of the PbI_2_, PbI_2_ residues in the perovskite film could lower the opportunities of charge carrier recombination, thus lead to an improved photocurrent (*I*_ph_) and EQE of PePDs [16,38].

To demonstrate the effect of the BPTCD additive upon device performances, the *I*_ph_ and *I*_d_ of PePDs were tested under white light irradiation and dark condition, respectively. The semi-log I-V characteristics of these PePDs are shown in Figure 2a and Appendix A. Typically, a low *I*_d_ is a key factor for high performance PePDs [12,13,16,40]. Here the control device with pure perovskite film shows a high *I*_d_ value of 2.93 × 10^−6^ A at −0.5 V, while the *I*_d_ for those PePDs with BPTCD additive reveal much lower values. The PePDs with 1, 2, 3 and 4 wt.% BPTCD show *I*_d_ values of 6.73 × 10^−8^ A, 1.06 × 10^−7^ A, 9.98 × 10^−8^ A and 7.06 × 10^−8^ A at −0.5 V, respectively. Nevertheless, the *I*_ph_ of the control device is slightly higher than that of the BPTCD added devices. The control device exhibits an *I*_ph_ of 4.74 × 10^−4^ A at −0.5 V while the *I*_ph_ of the devices with 1−4 wt% BPTCD additive are 8.75 × 10^−5^ A, 1.78 × 10^−4^ A, 2.33 × 10^−4^ A and 5.63 × 10^−5^ A at −0.5 V respectively. Since the most important parameter for PePDs is D*, which is taken into account the contribution from both photocurrent (on current) and dark current (off current). The on/off current ratio of the devices is studied as shown in Figure 2b and Appendix A. It can be seen that BPTCD added devices have much higher on/off current ratio (over 10^3^ at −0.5 V) than the control device. In addition, when the concentration of BPTCD additive increases to 4 wt.%, the on/off current ratio decreases significantly. Although the device with 4 wt.% BPTCD additive shows a relatively low *I*_d_, its *I*_ph_ is much lower than other devices as well.

To further investigate the effect of the different concentration of BTPCD additives on photo-gain, the EQEs of the PePDs were calculated and analyzed. As displayed in Figure 2c and Appendix A, the control device shows a relatively higher EQE value exceeding 40 % in the Visible region, which is benefited from a better device performance under light condition proved by the above results of I–V characteristics. The BPTCD added devices also exhibit efficient photo response in the Visible region. The EQE values of devices slightly increase with the increase of BPTCD concentration from 1 wt.% to 3 wt.%, and then decrease when the concentration is enhanced to 4 wt.%. Furthermore, according to the EQE spectra of the PePDs, we can estimate the D* of the devices over the Visible region based on the equations as follows:(1)$$R=EQE\times \frac{q\lambda }{hc}$$
(2)$$D^{*}=\frac{R}{2{\left(qJ_{d}\right)}^{\frac{1}{2}}}=EQE\times \frac{\lambda }{1240{\left(2qJ_{d}\right)}^{1/2}}$$
where *c* is the speed of light in a vacuum, *q* is the elementary charge of the electron. The calculated D* results at a bias of −0.1 V are shown in Figure 2d and Appendix A. Apparently, compared to the control device, the BPTCD added devices exhibit overwhelming D* values exceeding 10^12^ Jones over the entire Visible region. In particular, the device with 3 wt.% BPTCD additive achieves the highest D* value of 4.55 × 10^12^ Jones at a wavelength of 685 nm, where the enhancement of D* is attributed to the suppressed of *I*_d_. In addition, the photo response of the white light in PePDs with 3 wt.% BPTCD as a function of time was measured and shown in Figure 2e. The photocurrent of these devices is consistent and repeatable, which indicated good stability of the device. The spike liked cure of the initial part of on current is due to the time precision limit of our test equipment. Meanwhile, the temporal photo-response of the PePDs with 3 wt.% BTPCD was also measured (Figure 2f). The rise and fall times are defined as the times for the transient current rising from 10% to 90% and decreasing from 90% to 10% of the maximum output current, respectively. The rise and fall times for the PePD with 3 wt.% BPTCD were approximately 850 ms and 800 ms, respectively, which was the detection limit of our equipment. The overall device performance of our PePDs with BPTCD and comparison with recent reported PePDs are summarized in Table 1 and Table 2.

To disclose the reason for the suppressed *I*_d_ in BPTCD added devices, the morphology of perovskite films with various concentrations of BPTCD additives was characterized by SEM (Figure 3 and Appendix A). The pure perovskite film exhibits a relatively poor morphology with uneven grain sizes and many interior pinholes. These defects of the pure perovskite film can lead to a high leak current of the PePDs under the dark condition [33,37], which is also proved by I-V curve shown in Figure 2. After introducing BPTCD additive, the morphology of perovskite film changes significantly: the grain sizes become smaller and much more compact, thus resulting in a dense perovskite film. More significantly, the 4 wt.% BPTCD added perovskite film exhibits the smallest perovskite grain size, and the highest film coverage ratio. The improvement of film density can reduce the leak current of the devices, responsible for a suppressed *I*_d_. Atomic force microscopy (AFM) was also employed to study the morphology of perovskite films. As displayed in Appendix A, the perovskite film with 3 wt.% BPTCD shows smaller grain size and smoother surface than the pure perovskite. Larger grain size of pure perovskite film may lead to larger undulation thus rougher surface than perovskite film with 3 wt.% BPTCD. Benefited from smooth surface of perovskite film with 3 wt.% BPTCD, the contact interface property between perovskite film and PCBM film can be improved to suppress the leak current, resulting in low dark current. However, as shown in Appendix A, photocurrents of the devices with BPTCD additives decrease. Especially in the case of 4% BPTCD, the photocurrent is decrease by nearly one order compared to the control device. This phenomenon could be attributed to the decrease of grain size of MAPbI_3_ microstructure that would also lower the performance of the PePDs under light irradiation by unexpected loss of photo-induced charge carriers [39]. Benefited from the small-sized perovskite grains for 4 wt.% BPTCD added device, *I*_d_ decreased obviously, however, the *I*_ph_ also decreases, thus cause a performance deterioration. Therefore, a scrupulous balance between the grain size and film density should be considered for achieving high performance PePDs.

To illuminate the molecular interaction between BPTCD and PbI_2_, FTIR measurement was employed. As reported by Bi et al. [38], the C=O group can interact with PbI_2_ and trigger heterogeneous nucleation of MAPbI_3_, thus improve the crystallinity of the perovskite film. Similar results were also reported by Peng et al. [42], such that the C=O groups are responsible for the passivation of perovskite film via Lewis-base electronic passivation of Pb^2+^ ions, which reduces trap density and enhances the morphology of perovskite film. Here the used BPTCD as an additive is rich of C=O groups, which may share the similar effect on the improvement of perovskite film quality. As displayed in Figure 4a, the stretching vibration of C=O groups in the pure BPTCD shows at 1749 cm^−1^, while it shifts to 1743 cm^−1^ with the addition of PbI_2_. This result indicates a weakening of the C=O bond derived from molecular interaction between BPTCD and the PbI_2_ precursor, which is consistent with the previous reports [38,42]. The weakening of the C=O bond is indicative of the formation of an intermediate BPTCD-PbI_2_ adduct, which can be expected to improve the crystallinity of the perovskite film [38,43]. Therefore, in this work, the BPTCD additive plays a role to favor the heterogeneous nucleation of the MAPbI_3_, lower the energy barrier of nucleation and consequently to generate a high density of nuclei, leading to a more compact and smaller sized MAPbI_3_ microstructure [38,43,44].

The interfacial properties in the photodetectors dominate the charge extraction and transportation, thus influence the performance of PePDs [37]. Here, the UPS measurement (with a He I of 21.2 eV) of perovskite films on the ITO substrates was carried out to characterize the change of energy levels of the perovskite films with and without BPTCD additive (Figure 4b). The high binding energy cutoff region is shown in the left panel, while the onset region is in the right panel. The perovskite films with and without BPTCD additive share nearly the same high binding energy cutoff and the work function is estimated to be 3.92 eV. However, the valence band minimum (VBM) position of the pure perovskite film occurs at 1.38 eV below Femi level (E_f_) while the VBM position of perovskite film with BPTCD additive shifts to 1.62 eV below E_f_, indicating a deeper VBM energy level (Figure 4c). This energy shift could lead to a mismatch of energy levels between the perovskite film and PEDOT:PSS (HOMO of 4.9 eV), which hinders the hole injection into the active layer due to the energy barrier and therefore suppress the *I*_d_. However, this unmatched energy level could lead to the suppress of *I*_ph_ as well.

To study the influence of the trap density in the perovskite film on the dark current, the electron-only devices (ITO/PCBM/MAPbI_3_/PCBM/Ag) were fabricated to analyze the trap density by using the space charge limited current (SCLC) method. The I-V curves of the devices with and without the BPTCD additive are shown in Figure 4d. The linear I-V plot indicates an Ohmic response at low bias, and the current increase nonlinearly when the bias voltage exceeds the trap-filled limit voltage (V*_TFL_*), demonstrating that all the available trap states are filled by the injected charge carriers [45]. The onset voltage V*_TFL_* is linearly proportional to the density of trap states *η*_t_, which follows as Equation (3):*V*_*TFL*_ = *e**η*_*t*_*L*^*2*^/*2**ε**ε*_*0*_(3)
where *e* is the elementary charge of the electron (*e* = 1.6 × 10^−19^ C), *L* is the perovskite film thickness, *ε* is the relative dielectric constant of MAPbI_3_ (here we use 32 [45]), *ε_0_* is the vacuum permittivity (*ε_0_* = 8.854 × 10^−12^ F/m), and *η_t_* is the trap state density of perovskite film. The V*_TFL_* of the MAPbI_3_ film with and without BPTCD additive can be identified in Figure 4d, respectively. The thickness of perovskite film with and without BPTCD additive is 431 nm and 397 nm, respectively, derived from the cross-section SEM images in Figure 3c,d. The electron trap state density of perovskite film with BPTCD additive is estimated as 1.01 × 10^16^ cm^−3^, which is much lower than the perovskite film without BPTCD additive (1.50 × 10^16^ cm^−3^). This decrease of trap density is due to the Lewis-base nature of the oxygen atoms in the C=O groups on BPTCD, and Lewis base material can passivate the defects induced by Pb^2+^ as the recombination centers on the surface and grain boundaries of perovskite films [42]. Therefore, dark leakage current originated from defect could be significantly lowered by the passivation of BPTCD additive.

To further investigate the influence of BPTCD additive on the electrical properties of PePDs, the impedance of the devices was measured by an impedance analyzer under dark condition. The measured electrical parameters of R_1_, R_2_, R_3_, C_1_, CPE_1_-T and CPE_1_-P are listed in Table 3 through the fitting curves. In this circuit model (Figure 5a inset), the R_1,_ R_2_, and R_3_ correspond to the device series resistance, the interfacial resistance, and recombination resistance, respectively [37]. From the corresponding Nyquist plots, the R_1_ of the pristine and BPTCD added PePDs are similar. The R_2_ of BPTCD added devices is about two folds larger than that of the pristine devices. This result accounts for the great reduction of the leak current of BPTCD added devices [40,46,47] as depicted in the I-V curves in Figure 2a. The compact morphology, low trap density and relativity unmatched energy level between the active layer and PEDOT:PSS layer are responsible for the enhancement of R_2_. Meanwhile, a minor decrease of R_3_ in the control device is observed, which suggests that the pristine perovskite film with larger grain size can suppress charge recombination more effectively under light condition, resulting in a higher EQE and *I*_ph_ of the control device [48].

Finally, to examine the contribution of BPTCD additive toward the stability of PePDs, we performed a long-term photocurrent measurement on the PePDs with and without BPTCD additive. As shown in Figure 5b, the *I*_ph_ of the PePD with BPTCD additive can maintain 85% of the original value after operating 400 s, while the control device only maintains 47% *I*_ph_. The enhancement of device stability is attributed to the high-quality perovskite film with decreased defects, since the defects at grain boundaries typical provide charge accumulation sites and infiltration pathways for water vapor in air [49]. This is responsible for the irreversible moisture-induced degradation of perovskite, thus degrades the long-term stability of the devices. As mentioned above, the perovskite film with BPTCD shows much lower tarp density than the pristine perovskite film, indicating more stable crystal structure and enhanced long-term stability of PePDs.

## 4. Conclusions

In summary, we developed a simple route for controlling the microstructure of MAPbI_3_ by using BPTCD as an additive, and realized high performance PePDs based on the well controlled MAPbI_3_ film. The results showed that the interaction between C=O groups in BPTCD and the PbI_2_ in perovskite precursors could favor the heterogeneous nucleation of the MAPbI_3_, thus lower the energy barrier of nucleation and facilitate the growth of MAPbI_3_ crystal structures. Moreover, the BPTCD additive could down-shift the VBM level of the perovskite film and therefore contributes to the reduction of *I*_d_. As a result, the *I*_d_ of the devices was significantly suppressed by nearly two orders of magnitude compared to the control device but kept *I*_ph_ almost unchanged. Hence, with an optimal concentration of BPTCD additive (3 wt.%), the PePD exhibits a high D* value exceeding 10^12^ Jones over the Visible region with a maximum D* value of 4.55 × 10^12^ Jones at 685 nm at a bias of −0.1 V. Moreover, the PePDs with 3 wt.% BPTCD shows much greater stability comparing to the control devices. This work demonstrates a facile and low-cost method for tuning the MAPbI_3_ microstructure to obtain high quality perovskite film and open a novel route to realize high performance PePDs.

## Figures and Tables

**Figure 1 micromachines-11-01090-f001:**
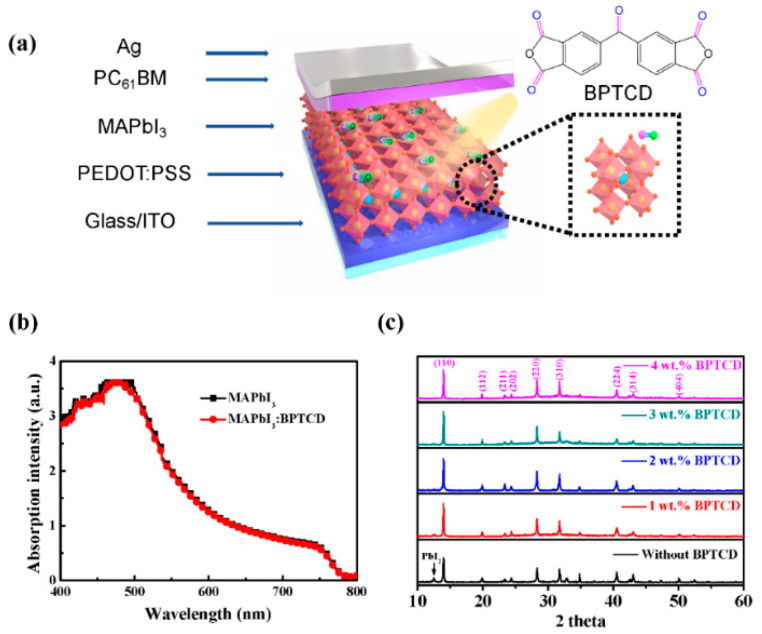
(**a**) A schematic diagram of the exploded view of PePDs. The dotted box shows the molecular structures of MAPbI_3_ and BPTCD. (**b**) Absorption spectra of the perovskite films with and without BPTCD additive (**c**) XRD patterns of the perovskite film with different concentrations of BPTCD additive.

**Figure 2 micromachines-11-01090-f002:**
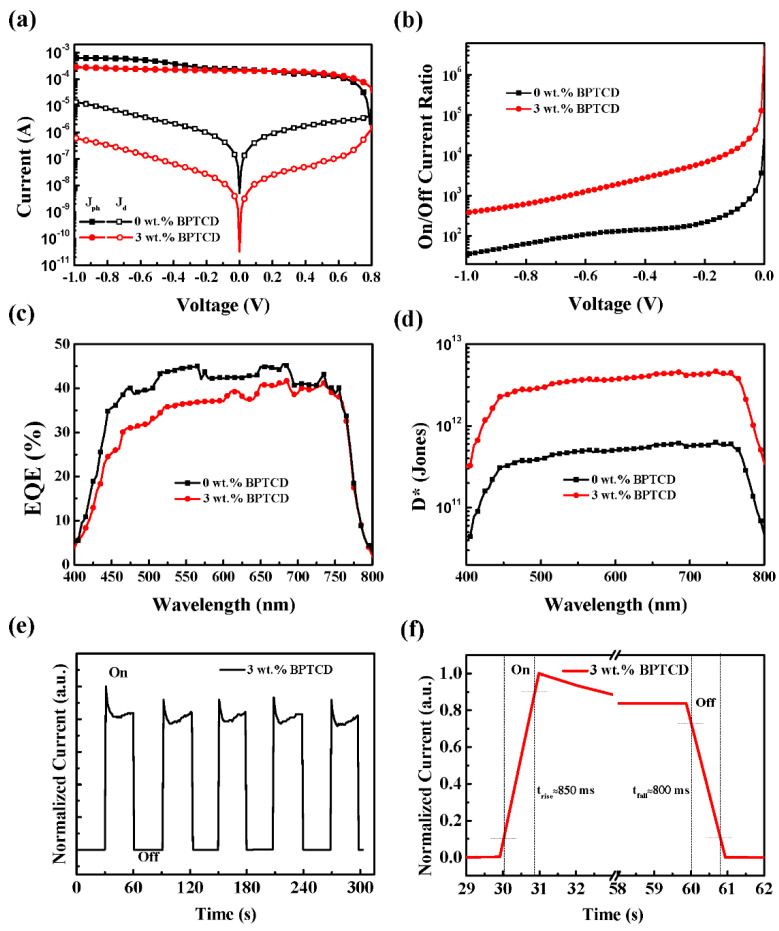
Electrical properties of PePDs with pure perovskite film and 3 wt.% BPTCD added perovskite film. (**a**) Semi-log I-V characteristics under light and dark condition; (**b**) On/off current ration; (**c**) Measured EQE spectra and (**d**) calculated D* values at −0.1 V; (**e**) Dynamic photo response measurements of PePDs with 3 wt.% BPTCD by multiple cycles introducing and removal white light (intensity of 100 mW/cm^2^); (**f**) Temporal photo-response of the PePDs with 3 wt.% BPTCD.

**Figure 3 micromachines-11-01090-f003:**
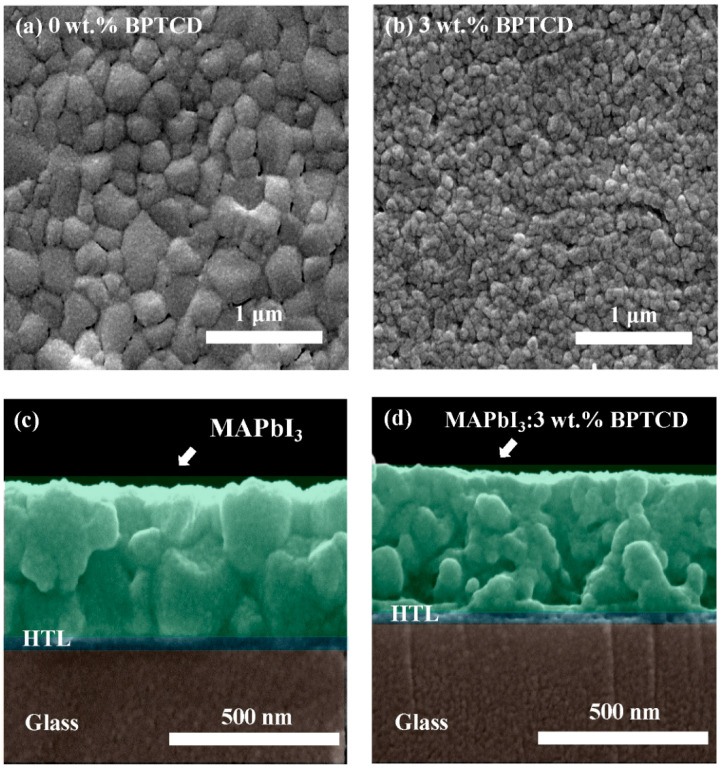
Top SEM images of perovskite films with (**a**) 0 and (**b**) 3 wt.% BPTCD additives. Cross SEM images of perovskite films with (**c**) 0 and (**d**) 3 wt.% BPTCD additives.

**Figure 4 micromachines-11-01090-f004:**
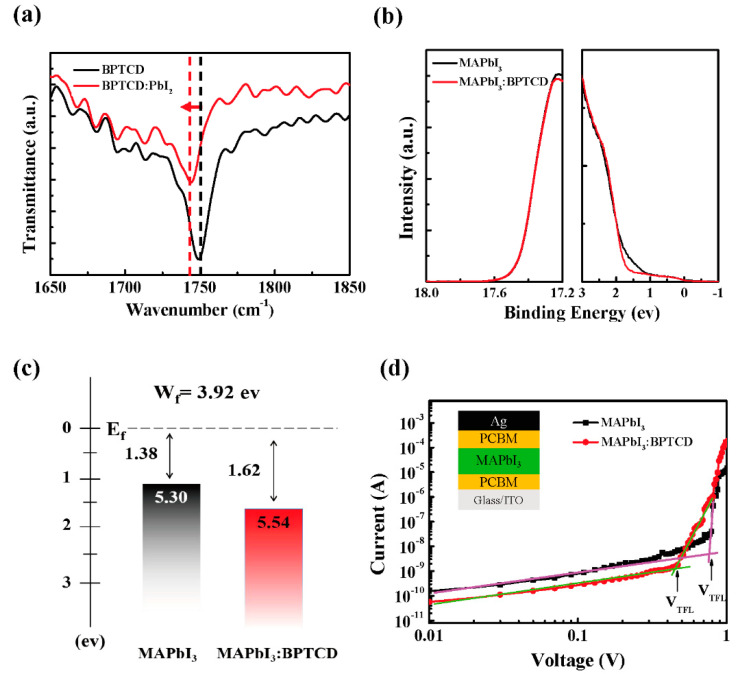
(**a**) FTIR spectra of pure BPTCD film and PbI_2_ added BPTCD film (**b**) UPS results and (**c**) Energy level diagram of perovskite films with 3 wt.% and without BPTCD additive (**d**) I-V characteristic of an electron-only device based on perovskite film with 3 wt.% BPTCD, with an inset showing the device structure.

**Figure 5 micromachines-11-01090-f005:**
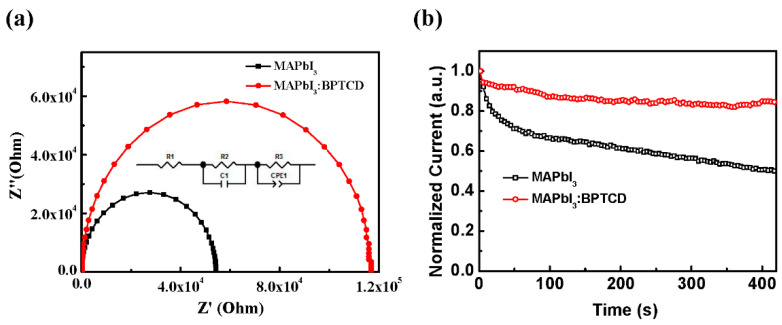
(**a**) Impedance spectra of PePDs, inset shows the corresponding equivalent circuit; (**b**) Long-term *I*_ph_ measurement of PePDs with and without BPTCD (under 100 mW/m^2^ white light).

**Table 1 micromachines-11-01090-t001:** Summarization of the *I*_d_, *I*_ph_ and on/off ratio of the PePDs at −0.5 V, and D* at 685 nm.

BPTCD	*I*_d_ (A)	*I*_ph_ (A)	On/Off Ratio	D* (Jones) @−0.1 V
Without BPTCD	2.93 × 10^−6^	4.74 × 10^−4^	0.12 × 10^3^	6.14 × 10^11^
1 wt.% BPTCD	6.73 × 10^−8^	8.75 × 10^−5^	1.20 × 10^3^	3.52 × 10^12^
2 wt.% BPTCD	1.06 × 10^−7^	1.78 × 10^−4^	1.29 × 10^3^	3.35 × 10^12^
3 wt.% BPTCD	9.98 × 10^−8^	2.33 × 10^−4^	1.86 × 10^3^	4.55 × 10^12^
4 wt.% BPTCD	7.06 × 10^−8^	5.63 × 10^−5^	0.63 × 10^3^	2.84 × 10^12^

**Table 2 micromachines-11-01090-t002:** Performance summary of reported perovskite-based photodetectors.

Materials	Detectivity (Jones)	EQE (%)	On/Off Ratio	Response Time	Ref.
Single-crystal CsPbBr_3_	6.2 × 10^10^@540 nm	-	1.5 × 10^3^	2.96 ms	[18]
Single-crystal MAPbBr_3`_	10^10^ in visible	-	10^4^	150 μs	[41]
MAPbI_3_	10^12^ in visible	80@ −0.1 V	-	280 ns	[22]
MAPbI_3_: PbS QD	10^12^ in visible, 10^11^ in NIR	38@ 0 V	10^3^	<500 ms	[13]
MAPbI_3_: BPTCD	4.5 × 10^12^ in visible	42@ 0 V	10^3^	≈800 ms	This work

**Table 3 micromachines-11-01090-t003:** Parameters of the equivalent circuit for PePDs with and without BPTCD additive.

Device	R_1_ (Ω)	R_2_ (kΩ)	R_3_(kΩ)	C_1_ (F)	CPE_1_-T (F/c^2^)	CPE_1_-P
Without BPTCD	28.4	57.2	1.5	2.18 × 10^−9^	2.07 × 10^−9^	0.91
With BPTCD	26.9	111.7	1.7	2.36 × 10^−9^	2.23 × 10^−9^	0.95

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
