# Peer review of "Additive Modulated Perovskite Microstructures for High Performance Photodetectors"

_micromachines, 2020, doi:10.3390/mi11121090_

Round 1
Reviewer 1 Report
This manuscript describes the optical properties of MAPbI3 perovskite according to the addition of BPTCD. I recommended publishing in micromachines journals because not only the recently popular perovskite material is used in this manuscript, but also a lot of effort is shown for the overall device characterization. However, if the following items are added or modified, it will be a better manuscript.
1. The abstract explained that porous nature causes high dark current. Why? And there seems to be no information related to porous except for the abstract, so it would be better to put the contents related to porosity in the main or to remove the contents about porous in the abstract.
2. (line 46, page 2) I think it is generally correct to write 'Vis' in the full name.
3. (line 89, page 2) It was said that the measurement was performed under white light. How did you get EQE or Detectivity? A description of this should be added to the 'materials and methods' section.
4. In most pictures, including Figure 1c, the text is too small to read.
5. (line 146, page 5) Figure 2a shows the current-voltage characteristics. On the other hand, J-V means current density-voltage. It is recommended to modify it with I-V.
6. In Figure 2e, the initial part of the 'on current' was bounced, like a spike. Why? Is this due to the noise or other factors?
7. There is no mention of the wavelength used in Figure 2e, f.
8. (line 214, page 7) What is the percentage of BPTCD added for the measurements of Figure 4?
Author Response
Summary Comment: This manuscript describes the optical properties of MAPbI3 perovskite according to the addition of BPTCD. I recommended publishing in micromachines journals because not only the recently popular perovskite material is used in this manuscript, but also a lot of effort is shown for the overall device characterization. However, if the following items are added or modified, it will be a better manuscript.
Our response: We thank the referee for these positive comments and suggestion of the revision. We carefully addressed the issues, as listed below, and we revised our manuscript accordingly.
Modifications: None.
Comment 1: The abstract explained that porous nature causes high dark current. Why? And there seems to be no information related to porous except for the abstract, so it would be better to put the contents related to porosity in the main or to remove the contents about porous in the abstract.
Our response: We thank the reviewer for this comment. Porous nature with unwanted crystallographic defects, such as pinholes and grain boundaries would cause not only abundant sites for charge-carrier recombination within the perovskite layer, but also a large leakage current in perovskite PDs, thus resulting in high dark current. Here we replace the term ‘porous’ with ‘unwanted crystallographic defects.
Modifications: In abstract Line 3, we replace the term ‘porous’ with unwanted crystallographic defects and modified the text as "However, the unwanted crystallographic defects of perovskites typically result in high dark current, and thus limits the performance of the device.".
Comment 2: (line 46, page 2) I think it is generally correct to write 'Vis' in the full name.
Our response: We thank the reviewer for this comment. We have displaced the 'Vis' by its full name 'Visible' in Line 50, Page 2. Moreover, we correct the 'Vis' throughout the manuscript.
Modifications: In Line 50, Page 2, Line 113, Page 3, 161, 163, 166, 174, Page 5, we replace the terms 'Vis' by 'Visible'.
Comment 3: (line 89, page 2) It was said that the measurement was performed under white light. How did you get EQE or Detectivity? A description of this should be added to the 'materials and methods' section.
Our response: We thank the reviewer for this comment. Quantum efficiency test system (Zolix SC 100) was used to obtain the EQE spectra then detectivity was calculated based on the obtained EQE results.
Modifications: In Line 93, Page 2, we added the texted as “Quantum efficiency test system (Zolix SC 100) was used to obtain the EQE spectra then detectivity was calculated based on the obtained EQE results.”.
Comment 4: In most pictures, including Figure 1c, the text is too small to read.
Our response: We thank the reviewer for this comment. We have modified the text in all the pictures in the manuscript.
Modifications: The front sizes in all the pictures are modified for better reading.
Comment 5: (line 146, page 5) Figure 2a shows the current-voltage characteristics. On the other hand, J-V means current density-voltage. It is recommended to modify it with I-V.
Our response: We thank the reviewer for this comment. We have modified the text “J-V” with “I-V” throughout the manuscript.
Modifications: In Line 137, Page 4, Line 154, Line 162, Page 5, Line 272, Page 8, we modified “J-V” with “I-V”.
Comment 6: In Figure 2e, the initial part of the 'on current' was bounced, like a spike. Why? Is this due to the noise or other factors?
Our response: We thank the reviewer for this comment. This is the original data of the dynamic photo response and the spike liked cure of the initial part of on current is due to the time precision limit of our test equipment. The curve of on current would be smoother with more precise test equipment.
Modifications: In Line 178, Page 6, we added the text as “The spike liked cure of the initial part of on current is due to the time precision limit of our test equipment.”.
Comment 7: There is no mention of the wavelength used in Figure 2e, f.
Our response: We thank the reviewer for this comment. The wavelength used in Figure 2e and 2f is white light. We have added the contents in Page 6, Line 176.
Modifications: In Line 176, Page 6, we modified the text as “In addition, the photo response of the white light in PePDs with 3 wt.% BPTCD as a function of time was measured and shown in Fig. 2e.”
Comment 8: (line 214, page 7) What is the percentage of BPTCD added for the measurements of Figure 4?
Our response: We thank the reviewer for this comment. For the measurement in Figure 4a, the concentration of the precursor solution was 50 mg/mL for the pure BPTCD sample. The precursor solution for the BPTCD:PbI2 sample was prepared by dissolving 25 mg BPTCD and 40 mg PbI2 in 1 mL DMF, as described in Method section. For the measurement in Figure 4b-4c, the percentage of BPTCD added is 3 wt.%.
Modifications: In Line 214 and Line 215, Page 7, we modified the text as “Energy level diagram of perovskite films with 3 wt.% and without BPTCD additive (d) I-V characteristic of an electron-only device based on perovskite film with 3 wt.% BPTCD, with an inset showing the device structure.”.
Reviewer 2 Report
The article is introduced using an additive method (BPTCD) for enhancement of the perovskite crystallization. In general, the manuscript is organized and has the potential to be published.
Line 3, I am not completely sure about the term ‘porous’ for the perovskite films.
The introduction needs to be extended by several more papers related to the surface modification of perovskite films. For example, these papers introduce different methods for surface treatment of perovskite films:
- https://doi.org/10.3390/en13153953
- https://doi.org/10.3390/met9090991
Figure 1(b), it seems to me that some part of the black graph is missed. I am not sure it happened during the submission/converting of the pdf…… Please check it.
Line 121, The claim is true also, it might be caused by decomposition induced by ambient exposure.
Line 130, ‘Typically, a low Id is a key factor for high-performance PePDs’ need a citation to a proper reference.
- Some of the discussions need to be extended such as XRD and specifically photocurrent.
- Are the smaller grains benefit the photodetector? Please make a discussion. Is it related to higher contact with PCBM and better movement of the carriers?
- The same concern about the grain size: the size of the grains is decreasing, how that can benefit the performance?
- Morphology: Is it possible to perform AFM measurement? Because from the SEM we could only see the size of the grains which substantially decrease after additive treatment.
.
Round 2
Reviewer 1 Report
I have checked the revised manuscript.
Most parts have been made clearer.
Therefore, I think that this manuscript can be published in micromachines.